# Stacking fault energy in concentrated alloys

Mulaine Shih [1], Jiashi Miao[1,2], Michael Mills[1,2] & Maryam Ghazisaeidi [1✉]

We revisit the meaning of stacking fault energy (SFE) and the assumptions of equilibrium dissociation of lattice dislocations in concentrated alloys. SFE is a unique value in pure metals. However, in alloys beyond the dilute limit, SFE has a distribution of values depending on the local atomic environment. Conventionally, the equilibrium distance between partial dislocations is determined by a balance between the repulsive elastic interaction between the partial dislocations and a unique value for SFE. This assumption is used to determine SFE from experimental measurements of dislocation splitting distances in metals and alloys, often contradicting computational predictions. We use atomistic simulations in a model NiCo alloy to study the dislocation dissociation process in a range of compositions with positive, zero, and negative average SFE and surprisingly observe a stable, finite splitting distance in all cases at low temperatures. We then compute the decorrelation stress and examine the balance of forces on the partial dislocations, considering the local effects on SFE, and observe that even the upper bound of SFE distribution alone cannot satisfy the force balance in some cases. Furthermore, we show that in concentrated solid solutions, the resisting force caused by interaction of dislocations with the local solute environment becomes a major force acting on partial dislocations. Here, we show that the presence of a high solute/dislocation interaction, which is not easy to measure and neglected in experimental measurements of SFE, renders the experimental values of SFE unreliable.

[1] Materials Science and Engineering, Ohio State University, Columbus, OH, USA. [2] Center for Electron Microscopy and Analysis, Ohio State University, Columbus, OH, USA. ✉email: ghazisaeidi.1@osu.edu

Stacking faults are irregularities in the sequence of crystalline planes. Therefore, a stacking fault in the ground state structure of a crystal is associated with an excess energy, called the stacking fault energy (SFE). SFE also measures the energy cost for shearing one atomic plane with respect to another, and as such is directly connected to the response of crystals to deformation. Stacking faults are created during the dissociation of lattice dislocations into partial dislocations to reduce the elastic energy according to the Frank's rule[1]. The size of the stacking fault region (distance between the partial dislocations) is thus determined by a balance between the repulsive elastic interaction between the partial dislocations and the energy to create the stacking fault between them, i.e., the SFE. In face-centered-cubic (fcc) crystals, SFE, and consequently the dissociation width of dislocations, is known to affect dislocation mobility, the ability to cross-slip and formation of twins, all of which govern the mechanical behavior[2–4].

Introducing chemistry change, through alloying, further affects the SFE and consequently the mechanical response. In a fcc crystal, the stacking fault region, bounded by partial dislocations, consists of two atomic planes with hexagonal-close-packed (hcp) structure. Suzuki et al. showed that the equilibrium concentration of solutes in this region can be different from the average bulk concentration[5,6]. The segregation or depletion of solutes to or from the stacking fault region, changes the SFE and further influences dislocation behavior. This phenomenon has been extensively observed in numerous alloy systems.

As the composition of alloys becomes more complex, for example in stainless steels or superalloys, alloying effects on SFE play a more prominent role in determining the competing deformation mechanisms[7–11]. For example, activation of secondary deformation modes in steels, such as martensitic transformation and mechanical twinning are directly related to the SFE. With decreasing SFE, the deformation mechanisms switch from dislocation glide to dislocation glide and twinning (Twinning-Induced-Plasticity or TWIP effect) to dislocation glide and $\gamma_{fcc}$ to $\epsilon_{hcp}$ martensitic transformation (Transformation-Induced-Plasticity or TRIP effect)[12–16].

High entropy alloys (HEAs) take the compositional complexity to a new extreme. HEAs are multicomponent alloys, in equal or near equal concentrations, where the notion of solutes and solvents breaks down[17–20]. In this case, the SFE is likely to be affected by local atomic configuration, as some atomic bonds are harder to break than others. Smith et al.[21] observed the local variations of stacking fault width along the dislocation lines in CoCrNiFeMn, proving the importance of local effects in HEAs.

The fundamental questions that follows are (1) can the SFE still be thought of as a unique intrinsic property of the crystal? and (2) are dissociation distances and dislocation mobility still governed by SFE?

These questions become more interesting in metastable alloys, where the fcc structure does not correspond to the lowest energy structure. For example, first principles calculations have shown that the equiatomic CrCoNi prefers a hcp structure at lower temperatures, even though it crystallizes in the fcc structure[22–25]. This immediately suggests that the creation of stacking faults in the fcc CrCoNi is energetically favorable. Calculations of SFE for a variety of atomic configurations show a negative average SFE and a wide spread due to local solute environments.

Dissociated dislocations in equiatomic CrCoNi medium entropy alloy are characterized using weak beam dark field scanning transmission electron microscopy (WB DF STEM)[26]. Details of the mechanical testing of equiatomic CrCoNi medium entropy alloy can be found in a previous publication[27]. Figure 1a shows an WB DF STEM image of a dissociated dislocation with a near edge character in CrCoNi. The image was acquired using a diffraction vector of $3g_{(\bar{2}02)}$. Under this imaging condition, both Shockley partial dislocations are visible, while the stacking fault on (111) plane is invisible. Figure 1b shows the measured dissociation distances between Shockley partial dislocations in equiatomic CrCoNi medium entropy alloy as a function of the characteristic angle of dislocations at both room temperature and cryogenic temperature conditions. There is no significant difference in dissociation distances between room temperature and cryogenic conditions. The dissociation distance $d$ is related to the dislocation characteristic angle ($\theta$) and SFE $\gamma$ via

$$d = \frac{K}{2\pi} \frac{\mathbf{b}_i(\theta)\mathbf{b}_j(\theta)}{\gamma} \qquad (1)$$

where $K$ is the elastic energy factor, from the sextic formalism of anisotropic elasticity and $\mathbf{b}_i(\theta)$ and $\mathbf{b}_j(\theta)$ are the Burgers vectors of $a/6\langle112\rangle$ Shockley partial dislocations oriented at angle $\theta$ with respect to the dislocation line[28]. Details of the anisotropic elasticity calculations and comparison with the isotropic elasticity formalism is presented in the Supplementary Note 1. The elastic constants used here are $C_{11} = 249.4$ GPa, $C_{12} = 159.0$ GPa, and $C_{44} = 138.4$ GPa from Laplanche et al.[29]. The SFE is determined by fitting the above equation to the experimental measurements of $d$. Based on the experimental results measured using WB DF STEM imaging, the SFE of equiatomic CrCoNi medium entropy alloy is estimated to be between 10 and 20 mJ/m². Importantly, while SFE measurements show a range of values, the average SFE is positive, which seems at odds with computational predictions. A similar measurement of SFE, using the isotropic elasticity formalism, has been presented by Laplanche et al.[30]. Addressing this apparent disagreement between computations and experiments is the main goal of this paper. Possible explanations offered so far are the effect of short-range ordering and temperature dependence of the SFE value in CrCoNi and CoCrNiFeMn[24,25,31–33]. These are plausible explanations. However, here we propose a more fundamental explanation that is not unique to the CrCoNi system and should be considered in all concentrated alloys. Namely, in transitioning from pure metals to concentrated alloys, we need to revisit the assumptions on equilibrium state of dislocations, since the dislocation/solute interaction typically not considered in measuring the dissociation distance simply cannot be ignored for concentrations beyond the dilute limit.

We demonstrate this computationally using a model NiCo system, which is fully miscible and allows for examining a range of compositions and temperatures. In addition, the hcp vs. fcc favorability, and consequently the sign of SFE, can be tuned by changing the composition. Moreover, this system is not prone to SRO formation, and as such allows for separating this effect from those caused merely by compositional fluctuations in a random alloy.

In the following chapters, we present our results on the relationship between SFE and equilibrium dissociation distance and show that the dislocation/solute interaction energy term cannot be neglected. Since this value is hard to measure and is often neglected during measurement of SFE, the experimental measurements are expected to overestimate the SFE. The lattice resistance to dislocation motion is almost negligible in pure fcc metals and increases as the concentration of solutes increase beyond the dilute limit. Therefore, in going from pure metals to alloys, the contribution of dislocation/solute interaction to the total energy becomes increasingly important.

## Results and discussion

**Choice of model alloys.** To investigate the deformation mechanisms in metastable fcc alloys, we use NiCo as our model

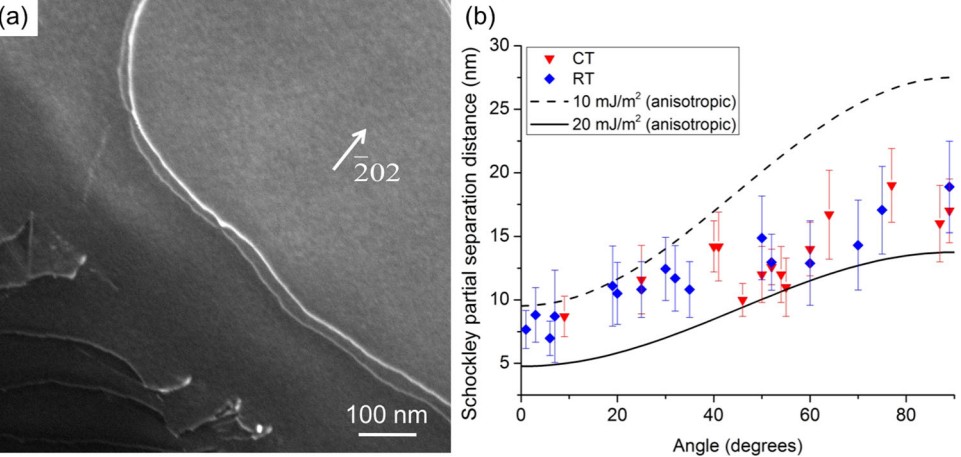

**Fig. 1 Characterization of dissociated dislocations in equiatomic CrCoNi medium entropy alloy. a** An example image showing both Shockley partial dislocations of a dissociated dislocation obtained by Weak beam dark field scanning transmission electron microscopy (WB DF STEM); and (**b**) experimentally measured dissociation distances in equiatomic CrCoNi medium entropy alloy at room temperature (RT) condition and cryogenic temperature (CT) condition.

computational system. Our rationale for this choice is as follows. First, the NiCo system is relatively simple and forms a solid solution across the entire concentration range. Therefore, we can span through the composition space without forming intermetallics. With pure Co (hcp) and pure Ni (fcc) on either end, NiCo is a good system for tuning the SFE. Second, short-range order (SRO) effect is negligible between Co and Ni. As such, we can ignore SRO effects and focus on the completely random fcc alloys. Third, we can find a reliable interatomic potential for MD simulations[34]. This potential yields hcp stable Co and fcc stable Ni. The relative stability of hcp and fcc phases, and consequently the SFE, can be tuned by alloying at various concentrations. Note that the trend of lowering SFE with the addition of Co is in line with previous experimental studies, but the corresponding concentrations are different[35,36]. We compared the energy/atom of fcc and hcp phases and chose three concentrations that represent positive ($Co_{70}Ni_{30}$), zero ($Co_{85}Ni_{15}$), and negative ($Co_{90}Ni_{10}$) SFE scenarios for the fcc alloy system.

We have calculated the free energy difference between the fcc and hcp phases ($F_{hcp} - F_{fcc}$) for the NiCo alloy (see below). The free energy difference for Ni has a negative slope, while for Co it has a positive slope. The trend is similar to previous DFT results, where Ni has a decrease in $F_{hcp} - F_{fcc}$, Co and NiCoCr alloys have an increase in $F_{hcp} - F_{fcc}$ as the temperature increases[22,23]. In addition to pure Ni and pure Co, free energy difference for each concentration is averaged over six random alloy configurations. The standard deviation between different configurations at temperature below 1300 K is <0.2 meV/atom. Thus the error bar is not visible in the figure. The $Co_{70}Ni_{30}, Co_{85}Ni_{15}$ and $Co_{90}Ni_{10}$ alloys were found to have little to no temperature dependence when the temperature is below 850 K. The free energy difference is about 3, 0, and −1 meV/atom for $Co_{70}Ni_{30}$, $Co_{85}Ni_{15}$, and $Co_{90}Ni_{10}$ alloys, respectively. The $F_{hcp} - F_{fcc}$ suggest that the mean SFE for the $Co_{70}Ni_{30}$, $Co_{85}Ni_{15}$, and $Co_{90}Ni_{10}$ alloys remain constant up to 850 K. A second-order SFE approximation using the free energy of fcc, hcp, and double hcp (dhcp) structures showed similar results for the alloys. More details can be found in Supplementary Note 2. We analyze the force balance using local SFE distribution at 78 K.

**Dissociation of an edge dislocation under equilibrium.** A dislocation with a full $1/2[1\bar{1}0]$ Burgers vector dissociates into two Shockley partial dislocations according to Frank's rule. This

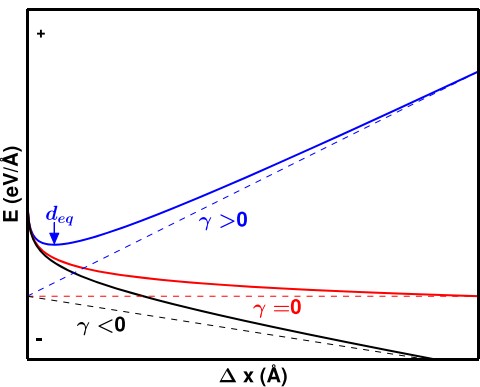

**Fig. 2 The schematic change in energy during the dissociation of a lattice dislocation.** Total energy (*E*) is shown as a function of the partial dislocation separation distance (Δ*x*) during the dissociation of a lattice dislocation into Shockley partial dislocations. The total energy is the sum of the interaction energy from the elastic strain around the dislocations, and the interaction energy from creating stacking fault. Here the blue, red and black solid lines correspond to the influence of positive, zero, and negative SFE. The sign of the SFE (*γ*) dominates the energy curve as the separation distance increases.

spontaneous process occurs in order to reduce the total elastic energy. The two partial dislocations repel one another while a positive SFE counteracts the repulsion. The equilibrium separation distance between the two partials corresponds to a minimum in energy (or zero force) as shown schematically in Fig. 2.

The situation is entirely different if the SFE is zero or negative as is the case in metastable alloys. When SFE is zero, only the elastic interactions remain with the repulsive force between the partials having an inverse relationship with their splitting distance. In case of the negative SFE, the partial dislocations are expected to dissociate infinitely since the forces from elastic interactions and SFE act in the same direction that push the dislocations apart. The energy vs. splitting distance curve monotonically decreases for zero and negative SFE alloys, and thus has no minimum for finite separation distances. This concept is illustrated as the black and red curves in Fig. 2. In other words, an equilibrium separation distance between the partial dislocations is not expected if the SFE is equal to or smaller than

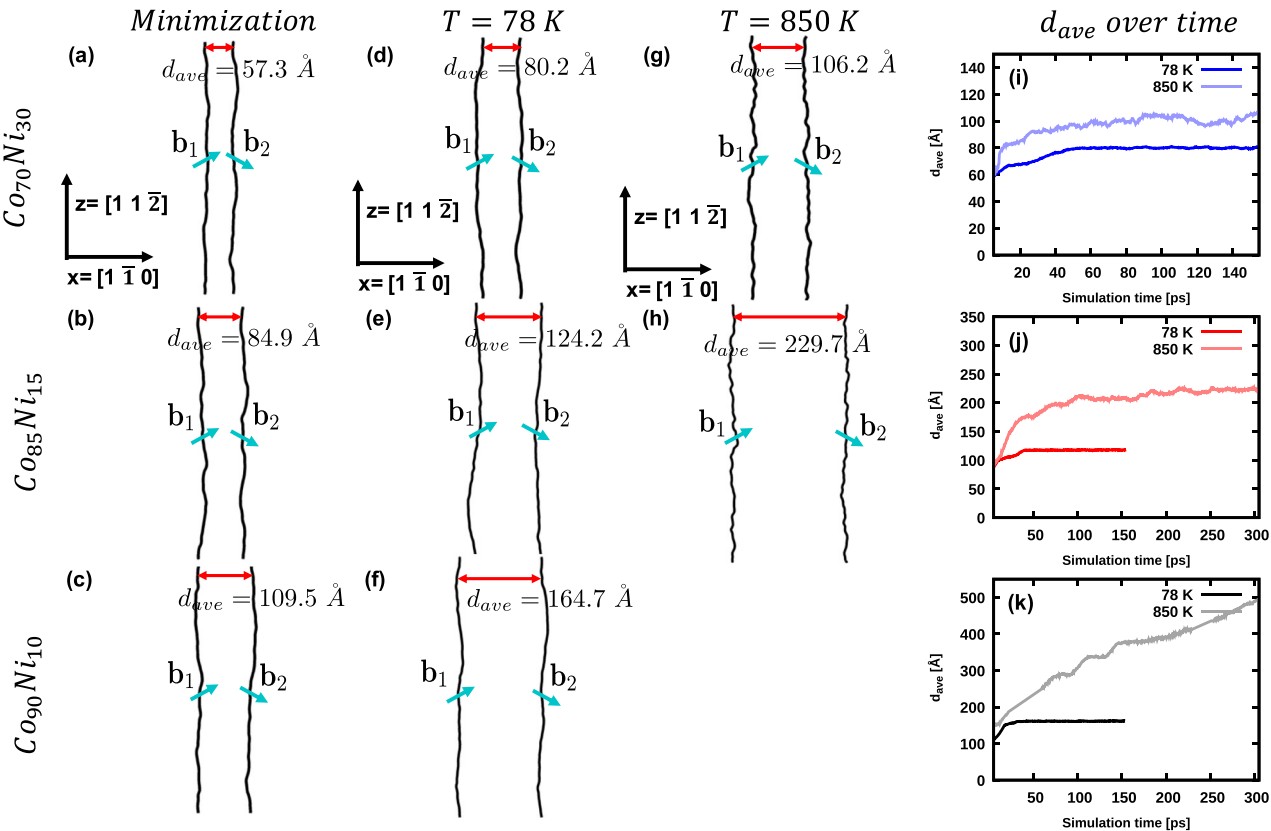

**Fig. 3 Dissociation of a 1/2[1$\bar{1}$0] edge dislocation in NiCo random alloys.** The equilibrium positions of the dissociated dislocation after energy minimization are shown in (**a**) $Co_{70}Ni_{30}$ (positive average SFE), (**b**) $Co_{85}Ni_{15}$ (zero average SFE), and (**c**) $Co_{90}Ni_{10}$ (negative average SFE); (**d–f**) show the dissociated edge dislocation at $T = 78$ K; (**g, h**) show the corresponding dislocations at $T = 850$ K. The dislocation analysis from OVITO is used, where the black lines represent the two partial dislocations (**b$_1$**, **b$_2$** are $\frac{a}{6}$<112>-type dislocations). Regardless of the sign of the average stacking fault energy values, the dissociation width of the edge dislocation remains finite at low temperature. **i–k** tracks the evolution of the average separation distance as a function of time, showing a continuous dissociation in $Co_{90}Ni_{10}$ alloy at $T = 850$ K.

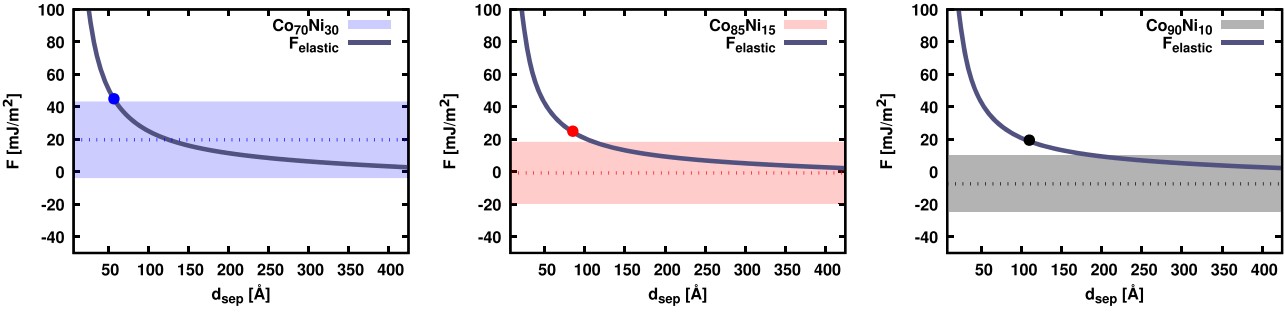

**Fig. 4 Forces acting on Shockley partial dislocations during the dissociation process.** The averaged SFE (dashed lines), local SFE (color blocks), and elastic interaction force per unit line (solid lines) are shown as a function of separation distance ($d_{sep}$). The local SFE blocks, which is plus/minus one standard deviation, can possibly intersect with the elastic interaction force curve as shown. Positive/zero/negative SFE are colored by blue/red/black, respectively. The circle dots are the averaged dislocation equilibrium separation distance from atomistic simulations after minimization.

zero. Note that a similar concept figure has been shown previously by Olson and Cohen[37].

Figure 3 shows the dissociation of the 1/2[1$\bar{1}$0] edge dislocation into Shockley partials in the three selected alloys with positive, near-zero, and negative SFE. In addition to energy minimization using conjugate gradient algorithm, we modeled the process at $T = 78$ K and $T = 850$ K using molecular dynamics (MD) to make sure that the dislocations are not trapped in shallow local minima of the energy landscape.

We use the dislocation analysis (DXA) implemented in OVITO[38] and only show the two Shockley partial dislocations.

The region in between the partial dislocations is the stacking fault. Each MD simulation ran for 155 to 305 ps depending on the convergence criterion explained in the Methods section. The averaged equilibrium separation distance ($d_{ave}$) is labeled on top of each dislocation and is obtained by averaging the separation distance along the dislocation line and over the last 50 ps of the simulation. The equilibrium separation distance between the partial dislocations remained finite in all three alloys (positive SFE: 57.3 Å, zero SFE: 84.9 Å and negative SFE: 109.5 Å) during energy minimization. The temperature dependence of $d_{ave}$ is as follows. The positive SFE alloy exhibits an equilibrium separation

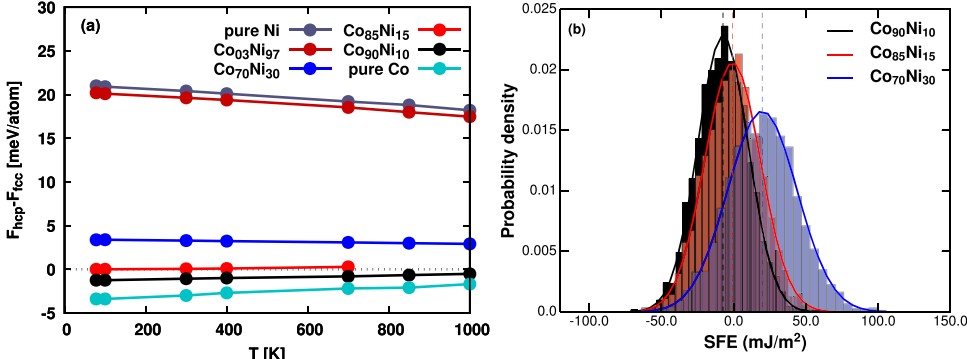

**Fig. 5 Comparison of finite temperature fcc-hcp free energy and local stacking fault energy in NiCo random alloy. a** The free energy difference between hcp and fcc phase for the NiCo alloy system. Negative energy difference indicates that the hcp phase is more favorable, and vice versa. $Co_{85}Ni_{15}$ (zero SFE) alloy calculations at temperature higher than or equal to 850 K have a strong tendency to become amorphous instead of forming stable hcp phase during TI. Results for zero SFE alloy is only considered up to 700 K. For the positive, zero, and negative SFE alloys, the energy difference is nearly constant at finite temperature, supporting the assumption that the mean of local SFE is constant across the simulation temperature range. **b** Normalized probability distribution of the local stacking fault energy (SFE) at finite temperatures for positive/zero/negative average SFE alloy at 78 K. The mean and standard deviation of the local SFE distribution is assumed as constant across the 1–850 K temperature range.

distance of 80.2 Å at 78 K and 106.2 Å at 850 K. The separation distance in the zero SFE alloy increases from 124.2 Å at $T = 78$ K to 229.7 Å at $T = 850$ K. The negative SFE alloy also has a finite equilibrium dissociation distance of 164.7 Å at $T = 78$ K, but continuously dissociated at $T = 850$ K. These results are surprising since the dislocation dissociation distances remained finite for the zero and negative SFE cases at low temperature, and in case of zero SFE alloy, even at elevated temperature.

Next, we compare the above splitting distances to the dimensions of our simulation box. Due to the periodic boundary condition in $x$ direction, we have an array of dislocations. We estimate the separation distance based on the elastic interaction between a partial dislocation with the other partial dislocation as well as all the periodic images. More details are presented in Supplementary Note 1. The elastic force on the partial dislocations is thus given by

$$F_{elastic} = K \frac{\mathbf{b}_1 \mathbf{b}_2}{2\pi} \left( \frac{1}{\Delta x} + \frac{-L_x + \pi \Delta x \cot(\pi \Delta x / L_x)}{L_x \Delta x} \right). \quad (2)$$

where $\mathbf{b}_1, \mathbf{b}_2$ are partial dislocations' Burgers vectors, $\Delta x$ is the separation distance between the partial dislocations and $L_x = 1100$ Å is the simulation box length in the $x$ direction. This equation is plotted in Fig. 4. In case of zero SFE, force balance is achieved only when $F_{elastic} = 0$. Therefore, the separation distance limit should be $L_x/2$. For zero and negative SFE we should expect the equilibrium separation distance to be equal and larger than $L_x/2 = 550$ Å, respectively. Except in the high temperature negative SFE case, the averaged equilibrium separation distances from our simulations are no larger than half of this limit. This unexpected finite dissociation is akin to the finite dissociation observed experimentally in Fig. 1 in CrCoNi, which also has a negative SFE based on DFT results[22,24].

Previous studies in concentrated solid solutions have shown that the separation distance between the partial dislocations vary along the dislocation line[21]. The variation is caused by local atomic configurations as certain bonds are easier/harder to break than others. Therefore, there is often a range of values associated with the SFE and even though the average value may be negative, the spread can span through both negative and positive values[22]. During the dissociation process in an alloy with negative average SFE, it is conceivable that one or both of the partial dislocations may encounter local regions of atomic distributions where bond breaking during slip results in a high energy cost. In other words, even with an average negative value, the local value might be large

enough to balance out the elastic repulsive force on the dislocation. The dislocation line tension, tends to keep the entire dislocation line together, resulting a finite dissociation distance between the partial dislocations. To test this hypothesis, we use the "local" SFE concept introduced in[21] to estimate the range of SFE values in selected alloys.

The detail of local SFE calculations is presented in the Methods section. Figure 5b shows the normalized frequency histogram, i.e., the probability distribution of local SFE values in the three selected alloys. The local SFE histogram fits closely to a Gaussian distribution function, an evidence of randomness in the model alloys. The mean and standard deviation of SFE values are obtained from this Gaussian distribution. The mean is averaged over the last 400,000 time steps of the simulation under NVT condition at $T = 78$ K. The mean value (average SFE) for $Co_{70}Ni_{30}$, $Co_{85}Ni_{15}$, and $Co_{90}Ni_{10}$ alloys are 19.8, −0.8, −7.4 mJ/m² at 78 K. The corresponding standard deviation are 24.0, 19.4, 17.5 mJ/m², respectively.

In addition, Fig. 5a shows that the energy difference between hcp and fcc phases is nearly independent of temperature in the model alloys. Therefore, we assume that the SFE values do not depend on temperature either.

Next, we estimate the effect of local SFE and compare with the elastic repulsive force per unit length $F_{elastic}$ between the partial dislocations. We have examined the effect of temperature on the elastic force between the dislocations by calculating the temperature dependence of the elastic constants. The details of these calculations are presented in Supplementary Note 2. Supplementary Fig. 2 confirms that $F_{elastic}$ does not change appreciably with temperature.

Figure 4 shows $F_{elastic}$, computed from elastic constants at $T = 0$ K, as a function of the splitting distance between the dislocations. Equation (1) implies that the intersection between the elastic force curve and the SFE value should represent the equilibrium condition. Since the SFE is no longer a unique value, we choose the average SFE plus and minus one standard deviation to represent the range of most likely SFE values. These values are represented by color blocks in Fig. 4.

The equilibrium $d_{sep}$ values from energy minimization simulations (Fig. 3a–c), are also shown on the same plot. In the positive SFE alloy, the equilibrium $d_{sep}$ coincides with the intersection of $F_{elastic}$ and the maximum value of the SFE. However, in the other two alloys, the equilibrium $d_{sep}$ is smaller than what is expected from the maximum SFE value. In addition,

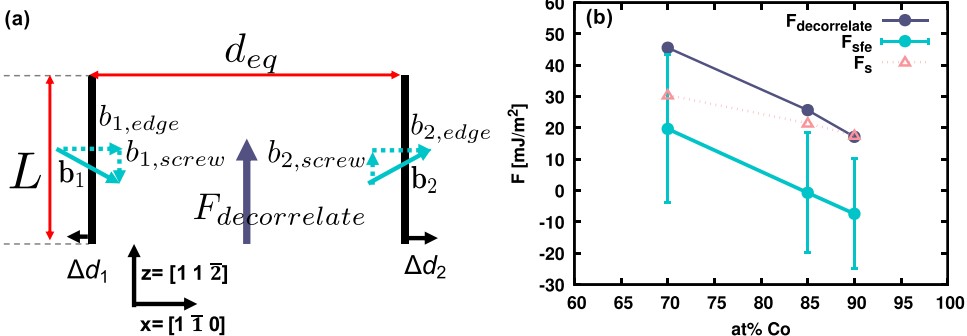

**Fig. 6 The decorrelation process of an edge dislocation in the NiCo random alloy. a** Schematic of a decorrelation process of two partial dislocations. The burgers vector of the two Shockley partial dislocations ($b_1$, $b_2$) can each be assembled by an edge and a screw component. A decorrelation force ($F_{decorrelate}$), acting parallel to the screw component, will drive the two partial dislocations to break away from equilibrium ($d_{eq}$) and move in opposite directions in the glide plane ($xz$). **b** The measured decorrelation force compared to the local SFE and force from solutes exerted on the partial dislocation for positive/zero/negative SFE alloys. The error bars represent one standard deviation in local SFE values.

an argument based on SFE alone cannot explain the temperature dependence of of dissociation distances in Fig. 3. The dissociation distances increase with temperature while the SFE does not change with temperature appreciably. The case of $Co_{90}Ni_{10}$ alloy is the most intriguing, in which the average SFE increases slightly with temperature, contrary to the trend in dissociation distance. These observations suggest the presence of another thermally activated process resisting the motion of the partial dislocations. The following section will examine the resistance to the glide of Shockley partials more closely.

**Examining the force balance on partial dislocations**. The conventional approach to dislocation dissociation based on pure fcc metals with negligibly low Peierls stress—assumes that the only forces acting on the partial dislocations are the elastic repulsive force per dislocation line length, exerted by the other partial dislocation, balanced by the energy per area cost for slip, i.e., SFE. Consider a dissociated dislocation in equilibrium. Assume, we can apply an external resolved shear stress that would push the partials to move in opposite directions. If the external shear stress exceeds the SFE, the partial dislocations will decorrelate and move away from one another, extending the stacking fault region in between. The smallest shear stress to achieve this should exert a force on the partials that is equal to the SFE. We call this critical value of shear stress the *decorrelation* stress.

In case of the dissociated edge dislocation considered here, the Shockley partials have equal and opposite screw components. Therefore, a shear stress resolved along the dislocation line direction (i.e., $\sigma_{yz}$), moves the dislocations apart. This process is illustrated in Fig. 6a schematically. We apply an initial $\sigma_{yz}$ stress starting from equilibrium configuration. We then incrementally increase the target $\sigma_{yz}$ stress by 40 MPa to find the critical decorrelation stress. The actual simulation shear stress is measured by averaging the global yz pressure output. The decorrelation force per unit dislocation line, is then obtained from $F_{decorrelate} = \sigma_{yz} \cdot b_s$, where $b_s$ is the screw component magnitude of the Burgers vector associated with the Shockley partial dislocation.

First, we calculate the decorrelation force in pure Ni and find the critical decorrelation stress to be 1730 MPa. This shear stress corresponds to a decorrelation force of 124 mJ/m². This value agrees well with the SFE of pure Ni which we calculate to be 125 mJ/m².

Next, we apply this method to the three fcc alloys. In case of alloys, the decorrelation stress depends on the dislocation line length. We performed a convergence test of the decorrelation

force vs. dislocation line length for different alloy concentrations and used the smallest line length beyond which the decorrelation stress remained constant. More details are presented in Supplementary Note 3. Supplementary Fig. 5 shows the decorrelation force for various dislocation line lengths in the three selected alloys. The decorrelation stress in the alloy, measures the local resistance to slip which should be temperature (and strain rate) dependent, if a thermally activated process contributes to this resistance. Here, we consider the maximum critical stress which is required in the absence of any thermal energy. Therefore, we have performed all the subsequent decorrelation stress calculations at $T = 1$ K.

Figure 6b shows the critical decorrelation force for the positive, zero, and negative SFE alloys compared against the SFE. The error bars represent the range of local SFE forces from the fitted Gaussian distribution mean and standard deviation. The upper bound for positive SFE alloy overlaps with the critical decorrelation force. In zero and negative SFE alloys, the decorrelation force is always higher than local SFE force. This indicates that there exists an additional force acting on the partials to achieve force balance in very low SFE alloys.

Given the fact that our model alloys are completely random solid solutions, the difference in the decorrelation force and local SFE, should therefore be due to the interaction of partial dislocations with local solute environments. For an accurate estimate of the solid solution strengthening, it is possible to use the analysis and theory developed by Varvenne et al.[39,40] with modifications for partial dislocations. If the energy barrier from all sorts of solute/dislocation interactions, including SRO, is of interest, it is also possible to use the analysis from Antillon et al.[41] to make reasonable estimations.

Next, we use the framework provided by the theory of solid solution strengthening of Varvenne et al.[40] to estimate the local potential energy barrier for solute–dislocation interactions. We emphasize that here, we study the dissociation of a perfect lattice dislocation under zero stress. Therefore, while the partial dislocations repel one another, the entire dislocation is not moving or bowing out and the roughness along the dislocation line is at a much smaller length scale than the amplitudes corresponding to the bowing segments due to solid solution strengthening. The solute/dislocation interaction is local and at the atomic scale. We are merely, borrowing the method used in Varvenne et al.[40] to approximate the local potential energy barrier, given by

$$\Delta E'_b(L_z, w) = \sqrt{2}\sigma_{\Delta U_{tot}}(L_z, w) \tag{3}$$

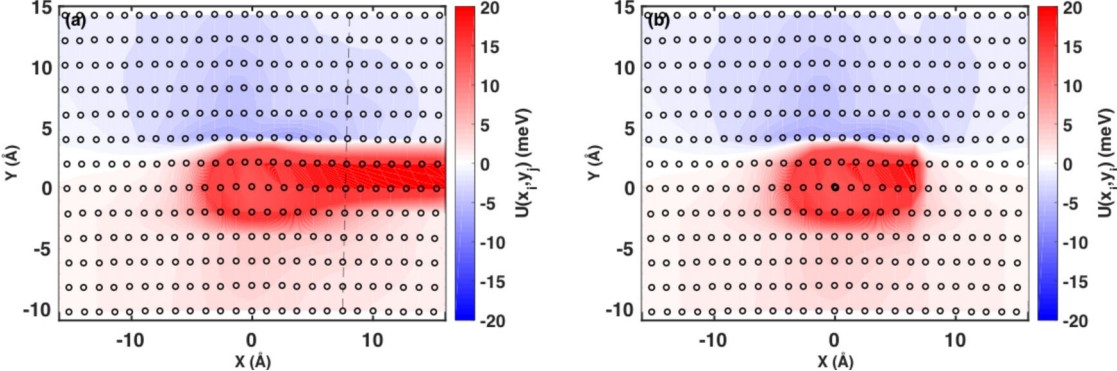

**Fig. 7 The Ni solute interaction energy map with a partial dislocation in fcc Co. a** Shows the calculated solute-dislocation interaction energy map, while (**b**) shows the modified solute-dislocation interaction energy map to eliminate the stacking fault interactions. We discard the interaction energies to the right of the dashed line and assigned the interaction energy to the sites on the right of the line such that the values are symmetric with respect to the partial dislocation center.

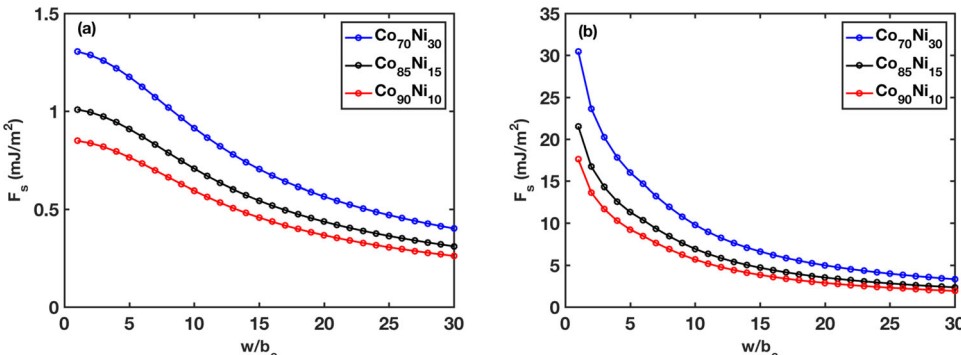

**Fig. 8 Estimation of solute/dislocation interaction.** The force per unit length of a partial dislocation exerted by solutes $F_s$ is computed as a function of glide distance from (**a**) reduced elastic interactions model and (**b**) direct solute–dislocation interaction energy.

where $\sigma_{\Delta U_{tot}}(L_z, w) = (\frac{L_z}{\sqrt{3}b})^{1/2} \Delta \tilde{E}_p(w)$ is the standard deviation of potential energy change from solute fluctuations and $w$ is the partial dislocation glide distance. As mentioned above, dislocation bow-outs and critical roughening amplitude—($w_c$) in ref. [40]—are not relevant here. Therefore, the potential energy barrier is a function of the dislocation line length ($L_z$) and the partial dislocation glide length $w$.

The energy barrier as a function of glide distance can be thought of as the work done by a force exerted from solutes on the dislocation, while the dislocation moves by a distance $w$. Therefore, for simplicity, we approximate the force per unit dislocation line, exerted by solutes as

$$F_s = \frac{\Delta E'_b(L_z, w)}{wL_z} \quad (4)$$

It is evident that $\tilde{E}_p(w)$ is the key quantity, which can be obtained by directly calculating the interaction energy map[39] of Ni solutes in fcc Co, or approximated by the reduced elastic interactions model[40]. Here, we take both approaches and compare the predictions.

First, we use the reduced elastic interactions model from Varvenne et al.

$$\Delta \tilde{E}_p(w) = \frac{\mu(1 + \nu)}{3\pi(1 - \nu)} \left[ \Sigma_{ij} \Delta f_{ij}(w) \right]^{1/2} \times \left[ \Sigma_n c_n \Delta \bar{V}_n^2 + \sigma_{\Delta V_n}^2 \right]^{1/2} \quad (5)$$

We modify the factor for dimensionless pressure field $f(x_i, y_j) = \Sigma_i \Delta b_e(x_i) \frac{y_j}{(x_i^2 + y_j^2)}$ for a Shockley partial, where we

consider the Burgers vector $b_e$ from the edge character contribution of the partial dislocation such that $\frac{\Delta b_e(x_i)}{b_e} = e^{-x_i^2/2\sigma^2}/\Sigma_{k=-\infty}^{+\infty} e^{-x_k^2/2\sigma^2}$, $x_i = nb_e$ with $n = 0, \pm 1, \pm 2...$ and $\sigma = 3b_e$. In the equation, $\Delta f_{ij}(w)$ is the change of the anisotropic pressure field due to glide length $w$, $c_n$ is the concentration for element $n$ in an $n$-component alloy, $\Delta \bar{V}_n = 3\bar{V}(\frac{a_n}{a} - 1)$ is the average misfit volume of solute $n$, $\sigma_{\Delta V_n}$ is the standard deviation due to local fluctuations, where $\bar{V}$ and $a$ are the alloy volume and lattice parameter, and $a_n$ is the lattice parameter for element $n$.

Assuming the standard deviation ($\sigma_{V_n}$) is negligible, the second term only depends on misfit volume of different alloy composition. Here we use the average lattice parameter at each concentration from minimization and we get $\Sigma_n c_n \Delta \bar{V}_n^2 = 0.00429$, $0.00258$, $0.00183$ Å$^6$ for Co$_{70}$Ni$_{30}$, Co$_{85}$Ni$_{15}$, and Co$_{90}$Ni$_{10}$ alloys, respectively. The results happen to be similar to lattice parameter calculations from the Vegard's law.

Moreover, we calculate the partial dislocation/solute interaction energy map using direct atomistic simulation via[40],

$$\Delta \tilde{E}_p(w) = \left[ \Sigma_{ijn} c_n((U^n(x_i - w, y_j) - U^n(x_i, y_j))^2 + \sigma_{\Delta U_{ij}^n}^2 \right]^{\frac{1}{2}} \quad (6)$$

where $U^n(x_i, y_j)$ is the interaction energy for a solute of type $n$ located at position $(x_i, y_j)$, $\sigma_{\Delta U_{ij}^n}$ the standard deviation due to distribution of local fluctuations along dislocation line direction $z_k$ and is negligible in our model.

Figure 7a shows the interaction energy map of Ni solutes, with one of the partial dislocations in fcc Co. The calculation details are described in Supplementary Note 4. Recall that we seek to separate the effect of solutes on the core of the Shockley partial, from that on the SF region already considered in previous sections. However, partial dislocations cannot exist without creating a SF in the crystal.

In order to obtain a hypothetically separate interaction energy map for the partial dislocation core, we cut off the interaction energy from the stacking fault region, delineated with a vertical line in Fig. 7a. The interaction energy values for this region are then assigned by mirroring the map with respect to the center of the dislocation core, determined by a Nye tensor analysis[42]. This modified interaction energy map is shown in Fig. 7b. Note that in this case, the partial dislocation will only move to the left ($x < 0$) during the dissociation process. The two partial dislocations are identical, therefore only one is considered.

Figure 8 compares the estimated force per dislocation line length from solute interactions $F_s$ obtained from the reduced elastic interaction and atomistic interaction energy map. The first glide step $w = b_e$ corresponds to the highest force, which determines the maximum resistance from solutes to dislocation motion. The reduced elastic interaction (misfit volume) model predicts the the maximum forces to be about 0.8–1.3 mJ/m², while the direct solute–dislocation interaction energy prediction lies in the range of 17–30 mJ/m² for the three compositions. Going back to Fig. 6b, we compare the $F_s$ estimated from Fig. 8b to $F_{decorrelate}$. It is evident that the $F_{decorrelate}$ is equal to $F_s$ in the negative SFE alloy and nearly equal in the zero SFE alloy. Given the simplifications used to calculate $F_s$, this agreement is surprisingly well. These results demonstrate that (a) considering only the size misfit is insufficient to describe the solute effects on the dislocation core, and (b) the dominant resisting force to motion of partial dislocations in the negative and zero average SFE alloys is the interaction of solutes with the partial dislocation core.

Therefore, going from elemental fcc metals to concentrated alloys, the assumptions of equilibrium dissociation of lattice dislocations, leading to Equation (1) should be revised. Figure 9 demonstrates this concept schematically. The elastic interaction between the Shockley partial dislocations $F_{elastic}$ extends the stacking fault area. The equilibrium separation distance corresponds to the configuration where $F_{elastic}$ faces the maximum resistance from the lattice. In elemental fcc metals, the Peierls stress is typically much smaller than the SFE. Therefore, at equilibrium, $F_{elastic} = \gamma$, leading to Equation (1). A concentrated alloy is different in at least two major ways. First, the SFE consists of a range of values with its upper bound represented by $\gamma^{max}$. In addition, motion of dislocations in a field of solutes is accompanied by solute interactions with the dislocation core. The maximum barrier imposed by solutes is overcome by a force per unit length $F_s$. The maximum of $\gamma^{max}$ and $F_s$ dominates the motion of Shockley partials. Therefore, at equilibrium

$$F_{elastic} = \mathrm{Max}(\gamma^{max}, F_s). \quad (7)$$

The results of our atomistic simulations support this conclusion. The equilibrium separations denoted by circles in Fig. 4 intersect the $F_{elastic}$ curves at values corresponding to the decorrelation forces computed independently in Fig. 6. Figure 6 also reveals that the decorrelation force in each alloy is equal to the maximum of $\gamma^{max}$ and $F_s$ which are computed separately.

We note that, in our model NiCo alloys, SFE variations, and solute/dislocation interaction are the only sources of resistance to dislocation motion. However, in application to real alloys

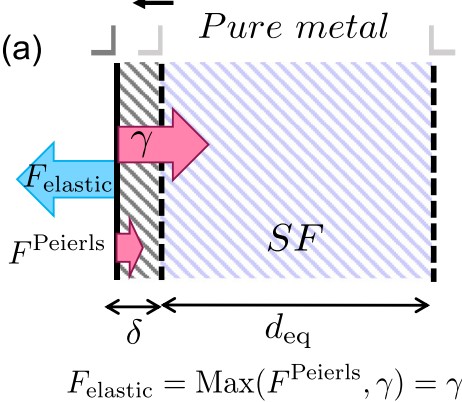

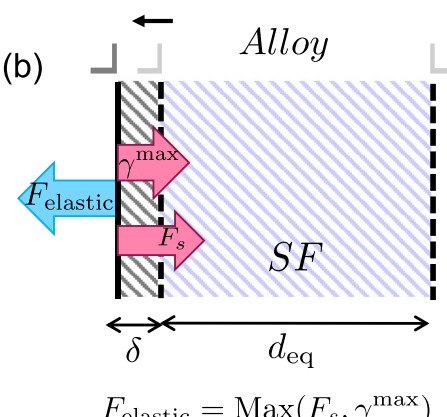

**Fig. 9 Schematic demonstration of the various forces, acting on a Shockley partial dislocation during dissociation.** The process is shown in (**a**) a pure metal and (**b**) an alloy. The elastic interaction $F_{elastic}$ pushes the left partial dislocation to the left, provided it can overcome the maximum resisting force to moving the dislocation incrementally by amount $\delta$. The Peierls barrier in elemental fcc metals is negligible hence the $F_{elastic}$ should counteract $\gamma$. In an alloy, SFE is a distribution with its upper bound represented by $\gamma^{max}$. Both $\gamma^{max}$ and the solute/dislocation interaction $F_s$ resist the motion of the Shockley partial. In order to move the dislocation by an amount $\delta$ beyond the equilibrium position, $F_{elastic}$ needs to counteract the maximum resistance given by the maximum of $\gamma^{max}$ and $F_s$.

other sources might be present. For example, in alloys with a tendency for SRO formation, the SRO contribution should be added in Eq. (7).

We have shown that in transitioning from pure metals to concentrated alloys, the assumptions on equilibrium dissociation of lattice dislocations must be revisited. Conventionally, equilibrium distance between partial dislocations is determined by a balance between the repulsive elastic interaction between the partial dislocations and the cost to create the stacking fault ribbon, quantified as SFE. This concept is valid in pure close-packed metals, where the SFE value is unique and the lattice resistance to motion of partial dislocations is small. However, this assumption is used to measure dissociation distances in alloys as well, an example of which is shown in Fig. 1. As the alloy concentration increases beyond dilute limit, the SFE represents a range of values and the dislocation/solute interaction becomes increasingly important. Ignoring the dislocation/solute interaction energy results in an overestimation of SFE and explains the discrepancy between experimental measurements and computational predictions of SFE based on DFT calculations. Figure 9 summarizes this concept schematically.

We used the NiCo model system to span through the entire composition range and chose alloys with positive, near-zero, and negative average SFE values. In addition, the NiCo system is not prone to ordering and as such, we were able to separate the effect of SRO formation. The SFE values did not show a significant temperature dependence either. We demonstrated that an edge dislocation dissociates into Shockley partials with a finite separation distance at low temperature, even when the average SFE is zero or negative, suggesting that a new force balance state must exist in these alloys. We demonstrated the presence of an additional resisting force by measuring the decorrelation force and comparing against the range of possible SFE values. We then showed that additional resistance is due to the interaction of solutes with partial dislocation cores in the model NiCo alloys. This additional force is necessary to construct the force balance according to Eq. (7), but is often neglected and not easy to measure in experiments. This is the fundamental reason that experimental measurements of SFE values should be taken cautiously, particularly in concentrated alloys.

## Methods

**Computations.** Atomistic simulations are performed using large-scale atomic/molecular massively parallel simulator[43]. We use the Kim et al. Modified Embedded Atom Method potential for NiCo alloy simulations[34]. The NiCo alloy is chosen as a model system to study the dislocation behavior in fcc concentrated solid solution alloys with a range of average stacking fault energies. Increasing the concentration of Co favors the hcp structure over fcc. Therefore, by varying Co concentration, we can access alloys that transition from stable fcc (positive SFE) to metastable fcc (negative SFE) values. The transition concentration for the alloys is roughly 85at% Co for the given potential.

First, we create random defect-free cells and calculate the equilibrium lattice constants at 78, 300, and 850 K under constant temperature and zero pressure (NPT) conditions. The thermal expansion ratio obtained from NPT simulations is used to expand and remap the atoms in the dislocation cell.

Simulation cells containing dislocations are oriented along $x = [1\bar{1}0]$, $y = [111]$ and $z = [\bar{1}\bar{1}2]$ directions respectively. We varied the dislocation line length ($L_z$) in 69, 519, 813, and 1195 Å to ensure our results are converged. The simulation cell size shown in Fig. 3 was chosen as $1100 \times 523 \times 519$ Å in $x$, $y$, and $z$ directions and consists of 27,213,840 atoms. We first introduce a full dislocation, where half of the upper simulation cell is displaced with the shortest lattice vector, called Burgers vector, with respect to the lower simulation cell. We apply the anisotropic displacement-field solution to create a $\frac{1}{2}\langle 110\rangle$-type dislocation, the dislocation line is designated along the z-direction. The elastic constants of the NiCo alloy at a given concentration is interpolated from pure Ni and pure Co zero-temperature elastic constants of the potential. Periodic boundary conditions in the glide plane of the edge dislocation ($x$ and $z$ directions), and shrink-wrapped boundary condition in the out of plane ($y$ direction) are used. Two atomic layers of atoms on each the top and bottom side along the nonperiodic boundary are set to be fixed, while the rest of the system is relaxed to its equilibrium positions using energy minimization.

The dislocation cells are then thermalized at 78, 300, and 850 K using a Nose–Hoover thermostat under constant temperature and volume (NVT) conditions. The dislocation cells are rescaled according to the thermal expansion ratio at the target temperature. Langevin dynamics coupled with the target temperature was first applied for 5 ps to pre-thermalize and reduce the time for convergence. The system is then switched to NVT conditions until convergence is achieved. The $d_{sep}$ values are averaged over 50 ps intervals. We consider the simulation to be converged when the difference between the last two evaluated average $d_{sep}$ values is <2%. We first examine the external stress-free conditions, to see what is the finite temperature effect on dislocation cells. We then apply the target constant stress on the dislocation cell. We use the constant traction boundary condition to apply the desired stress on the dislocation. This method has been widely used in the literature[41,44]. We first strain the whole simulation cell according to our target stress state. Then we apply the force $F_j = \pm \sigma_{ij}A/N$ on each atom in the top and bottom layers. To study the decorrelation force on the partial dislocations, we apply various $\sigma_{yz}$ values and obtain the critical value. The same conditions were used, where Langevin dynamics was first applied and then switched to NVT conditions for 100–150 ps.

We use thermodynamic integration (TI) to calculate the free energy of fcc and hcp phases of the chosen NiCo alloys. Given a known initial equilibrium state, we can numerically integrate the work done to reach another equilibrium state of interest. In this study, we have employed the nonequilibrium TI method developed by Freitas et al.[45]. We get the free energy of the final equilibrium state by adding the free energy difference such that

$$F_f(N, V, T) = F_E(N, V, T) + \Delta F. \tag{8}$$

Where $F_f(N, V, T)$ is the free energy of our final equilibrium state, $F_E(N, V, T) = 3Nk_BT\ln\left(\frac{\hbar\omega}{k_BT}\right)$ is the reference equilibrium state where we use the known Einstein crystal, and $\Delta F$ is the free energy difference. The nonequilibrium TI method assumes that the forward and backward integration processes are identical. Therefore we can approximate the change in free energy as

$$\Delta F = \frac{1}{2}\left[W_{irr, E\to f} + W_{irr, f\to E}\right]. \tag{9}$$

Where $W_{irr, i\to j}$ is the irreversible work done to transform from state i to state j. In our study, we use 32,928 atoms and six different random configurations for the alloys to compute the free energy of both fcc and hcp phases.

To study the effect of local SFE distribution, a large cell with 480,000 atoms is used. In which we assign atoms into 2500 small groups, each one having $\approx 87$ Å$^2$ SF area, so that we have the mean and standard deviation that are statistically significant. A tilt method for local and average SFE calculations as a function of concentration and temperature are performed[8]. The bulk reference cells are first constructed and minimized along $[1\bar{1}0]$, $[11\bar{2}]$, and $[111]$ direction, where the Ni and Co atoms are dispersed randomly according to the given concentration. A translation vector is then applied to the perfect cell to get the SF cell structure, and minimization is performed with $x$, $y$ positions fixed. The zero-temperature SFE is computed using the difference in energy of the two cells divided by the SF area. The finite temperature SFE are computed similarly. We apply NVT simulations at 78 K after rescaling the box according to thermal expansion ratio and remapping the atoms.

**Experiments.** Experimental details regarding to the fabrication of test alloy, general microstructure, and deformation behavior can be found in a previous publication[27]. Specimens for transmission electron microscopy (TEM) study were mechanically ground down to a thickness of about 100 μm. Final perforation of TEM foils was completed using electropolishing in an electrolyte consisting of 20% perchloric acid in methanol at −30 Å °C and a voltage of 10–13 V. Dissociated dislocations were characterized using the recently developed WB DF STEM[26]. The details of WB DF STEM imaging can be found elsewhere[26]. WB DF STEM imaging was conducted in a FEI Tecnai TF20 TEM/STEM microscope operating at an accelerating voltage of 200 kV.

## Data availability

All relevant data are available upon request from the authors.

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

## Acknowledgements

We acknowledge the support of this work by the National Science Foundation Grants DMR-1553355 and DMR-1905748. Computational resources were provided by the Ohio Supercomputer Center.

## Author contributions

M.S. performed the atomistic simulations and wrote the draft with input from all authors. M.G. designed the research based on discussions with M.M. M.S., M.M. and M.G. analyzed the results. J.M. performed WB DF STEM and analysis. All authors discussed the results.

## Competing interests

The authors declare no competing interests.
