## [Peer Review File · Nature Communications]

REVIEWER COMMENTS

Reviewer #1 (Remarks to the Author):

The paper "Stacking fault energy in concentrated alloys" presents a new perspective on how to interpret stacking fault energy (SFE) measurements based on partial separation in concentrated alloys. Given the current broad interest in concentrated alloys/high entropy alloys field and the importance of SFE to determine their mechanical properties, this work is very relevant to the materials community.

1. Here, the discrepancy between SFE calculations and experimental determination of the SFE based on partial separation is interpreted in terms of an extra term that enters the force balance that determines the partial separation, which is the friction force due to the interaction between solutes and partial dislocations. The main questions I would like to ask the authors is whether they envision this force to be temperature and strain-rate dependent, as it should be when thinking of "solute-strengthening" (due to the solute-dislocation interactions), or rather some constant term. Did the authors verify that the partials, in their simulations at constant applied stress, do not move any further after 100-150 ps? How was this relaxation time chosen? I am wondering whether a sufficiently large relaxation time can be found, such that the partial move until the separation distance determined by the sole elastic force and alloy SFE (equation 3). Then, the friction force should be zero for sufficiently large timescales – which is probably more relevant than ps timescales when considering experiments. If the friction contribution to the decorrelation force is time-dependent, then the discrepancy between $F_{decorrelate}$ and F_{sfe} in Figure 6 does not look surprising to me.

Along with this main question, I have some few more:

2. Authors mention that there is no significant difference in dissociation distances between room and cryogenic temperature condition. Have you verified that above RT the dissociation distance also does not change?

3. What is the anisotropy of CrCoNi? How would d change by considering the Poisson's ratio and shear moduli on the crystallographic plane of the dislocation? And how would $F_{elastic}$ (Eq. 2) change?

4. While introducing the dislocation and relaxing the cells, which minimization procedure do the authors follow? Experience with this kind of simulations tells that not all methods (e.g. the standard conjugate gradient) converge to the correct separation distance.

5. When relaxing the cells, are authors sure that choosing 50 ps is enough for the system to achieve the equilibrium distance? What are the typical barriers due to the solutes for the glide of the partials? This would give an indication of how long one should wait to see things move, at a given temperature, and probably explain why "the dislocation separation distance remains finite at 850K, even for the negative SFE case".

Small typos:

Throughout the paper: Franks rule -> Frank's rule

Page 3: These questions becomes -> These questions become

Page 7: "staking fault" -> stacking fault

Reviewer #2 (Remarks to the Author):

RE: NCOMMS-20-23945

Stacking fault energy in concentrated alloys

By: Mulaine Shih, Jiashi Miao, Michael Mills, and Maryam Ghazisaeidi

The manuscript tries to answer the question of the legitimacy of using average SFE as a representative quantity for describing the dislocation structures in HEA. To this end, the authors did a careful study of the local chemical environment's effect on the structure of the dislocations in NiCo alloy. However, I have some concerns about the publication of this paper in NCOMMS at its current form.

1- The authors only used the isotropic elasticity as their theoretical framework. I strongly suggest the authors modify their formulation to the case of anisotropic elasticity. If it is not possible to find the close form solution for the separation distance versus the character angle, they should at least try to use anisotropic elasticity for screw and edge cases and discuss the possible effects.

2- The authors didn't consider the temperature dependence of the elastic moduli in this work. I guess at 850K the elastic moduli are considerably less than their 0K values. This decrease will change F_{elastic} . For coming up any sound conclusion about the separation distance, we need to analyze the whole temperature-dependent parameters. Sidenote: Even the Burgers vector magnitude will increase by increasing the temperature, but I guess its effect would be minimal. The anisotropy effect may intensify/decrease by increasing the temperature.

3-For Eq. 2, the authors only considered the five nearest dislocations. I guess one can include all the neighbors and find the close form of the F_{elas} and then use it in Fig. 5. The series looks convergent.

4- The authors used the difference in the free energy of hcp and fcc phases to prove that the SFE remains almost constant in the interest temperature domain. This approximation is fine. However, one can improve it by using the hcp double-layer energies [1]. I suggest the authors consider this improvement in their approximation at least for some representative samples.

5- The notion of the friction force is vague. Given the complete randomness of the system, one probably could quantify this friction force in terms of the fluctuations in the atoms' spatial distributions. This analysis should be the same as Varvenne et al. model for solute strengthening in HEA. The difference would be that now we have multiple partial dislocations repelling each other instead of the one dislocation under a resolved shear.

After implementing the major revisions suggested here, I think manuscriptmanuscript would be suitable for the publication in NCOMMS.

[1] Zhang, Xi, et al. "Temperature dependence of the stacking-fault Gibbs energy for Al, Cu, and Ni." *Physical Review B* 98.22 (2018): 224106.

RESPONSE TO REVIEWER COMMENTS

Reviewer #1:

Comment 0: “The paper “Stacking fault energy in concentrated alloys” presents a new perspective on how to interpret stacking fault energy (SFE) measurements based on partial separation in concentrated alloys. Given the current broad interest in concentrated alloys/high entropy alloys field and the importance of SFE to determine their mechanical properties, this work is very relevant to the materials community.”

Response: We thank the reviewer for a careful review of our manuscript and helpful suggestions.

Comment 1. “Here, the discrepancy between SFE calculations and experimental determination of the SFE based on partial separation is interpreted in terms of an extra term that enters the force balance that determines the partial separation, which is the friction force due to the interaction between solutes and partial dislocations. The main questions I would like to ask the authors is whether they envision this force to be temperature and strain-rate dependent, as it should be when thinking of “solute-strengthening” (due to the solute-dislocation interactions), or rather some constant term.”

Response: As the reviewer pointed out, solid solution strengthening is a thermally activated process, therefore it is temperature and strain rate dependent.

We intentionally called the extra term that enters the force balance, “friction force” to keep it more general than solute strengthening. In an arbitrary system, there could be various sources for this force, e.g. Peierls stress, SRO effect in addition to local solute effects. [we have added a new schematic figure 7 and some text on page 12 to clarify this further].

In our completely random simulations, in the absence of any other interaction, we conclude that this extra term must be due to solute interactions. However, our aim is not to compute the solute effect on the partial dislocations, rather to show that it exists!

Perhaps, the use of solute *strengthening* in this case, can be misleading. What we are studying is the dissociation of a perfect lattice dislocation under zero stress. Therefore, while the partial dislocations repel one another, the entire dislocation is not moving. The dislocation is not bowing out and the roughness along the dislocation line is at a much smaller length scale than the amplitudes corresponding to the bowing segments due to solid solution strengthening. The solute/dislocation interaction is local and at the atomic scale.

One could argue that the process by which the dislocation can escape from the local barrier of solutes is thermally activated. Thermal energy can overcome some of that energy barrier. In fact, the negative SFE alloy in Figure 3 shows this. The finite dissociation distance in this alloy increases with temperature, even though the SFE is not decreasing [c.f Figure 4 (a)].

Regarding the strain rate dependence, although it is irrelevant here for the equilibrium dissociation, it should have an effect during deformation, but in that case the dominant phenomenon becomes the solute strengthening of the whole dislocation at a larger length scale.

Comment 1, cont: “Did the authors verify that the partials, in their simulations at constant applied stress, do not move any further after 100-150 ps? How was this relaxation time chosen? I am wondering whether a sufficiently large relaxation time can be found, such that the partial move until the separation distance determined by the sole elastic force and alloy SFE (equation 3). Then, the friction force should be zero for sufficiently large timescales – which is probably more relevant than ps timescales when considering experiments. If the friction contribution to the decorrelation force is time-dependent, then the discrepancy between $F_{decorrelate}$ and F_{sfe} in Figure 6 does not look surprising to me.”

Response: The key here is the definition of the critical stress for decorrelation, which is the stress required to overcome the *highest* energy barrier in dislocation/solute interaction. Also, the critical stress is computed at $T=1K$. So, there is virtually no thermal energy.

Even if we assume that this critical stress is not exact, the mere comparison between pure Ni and the alloys proves our point that in case of alloys there should be an extra term! What the reviewer is describing is a thermally activated process that is absent in pure Ni.

At subcritical stresses and finite temperature, the dislocation in the alloys may at some point overcome its local energy barrier but it will move until it gets trapped in the next energy well. Above the critical stress, the dislocation has enough energy to overcome all barriers and therefore continues to move.

The dislocation/solute interaction will never become zero as long as dislocations encounter solutes. We incrementally increase the stress until the partials continuously move away and can overcome the highest barrier.

Below is an example of the dissociations between partials under subcritical stress and the distance at the critical value.

At sub-critical stress (5[MPa] below the critical stress), the partial dislocations do not move (ie. $d_{ave} = \text{constant}$) even if we run ~ 180 ps in total (take Ni₈₅Co₁₅ for example):

On the other hand, at the critical stress, the dislocations start moving continuously within the simulation time.

Comment 2: “Along with this main question, I have some few more: Authors mention that there is no significant difference in dissociation distances between room and cryogenic temperature condition. Have you verified that above RT the dissociation distance also does not change?”

Response: That is a great point. However, due to the limitation of the instruments, we only checked dissociation distance at room temperature and cryogenic condition and did not check dissociation distance above the room temperature.

Comment 3: “What is the anisotropy of CrCoNi? How would d change by considering the Poisson’s ratio and shear moduli on the crystallographic plane of the dislocation? And how would F_{elastic} (Eq. 2) change?”

Response: We thank the reviewer for raising this important issue. We have redone our analysis using anisotropic elasticity.

From the literature, the Zener anisotropy ratio for CrCoNi is about 3.06 (elastic constants from Laplanche et al 2020). We have modified equation 1 and added more details in supplementary note 1 accordingly.

We have also updated Figure 1.b by adding the predicted dissociation distances from anisotropic elasticity formulation. We kept the isotropic solutions for comparison in supplementary note 1 line 37, since that is the one used in literature. The difference is more pronounced for the edge segments, but still, the major conclusion of a positive, low average SFE is unchanged.

Comment 4: “While introducing the dislocation and relaxing the cells, which minimization procedure do the authors follow? Experience with this kind of simulations tells that not all methods (e.g. the standard conjugate gradient) converge to the correct separation distance.”

Response: We completely agree with the reviewer. After introducing the dislocation, we started with the conjugate gradient method for minimization since the convergence is faster than the steepest descent method. The concern raised by the reviewer is precisely why we ran dynamic simulations subsequently so that if the dislocation is trapped in a local shallow well it can be untrapped by thermal energy. We ran all the simulations at the target temperature and zero applied stress for 150 ps. The final equilibrium separation distances are averaged over the very last 50 ps of configurations of the simulations.

Comment 5: “When relaxing the cells, are authors sure that choosing 50 ps is enough for the system to achieve the equilibrium distance? What are the typical barriers due to the solutes for the glide of the partials? This would give an indication of how long one should wait to see things move, at a given temperature, and probably explain why “the dislocation separation distance remains finite at 850K, even for the negative SFE case”.

Response: The equilibrium state was determined when the total energy did not decrease any further and fluctuated around an average value. We actually ran the relaxations for 150 ps and the equilibrium distance is an average of the last 50 ps of simulations, as shown in the figures below. We believe the time is long enough for the system to find a local/metastable energy state. Figures below are cases for theCo90Ni10, 78 K and 850 K, the total energy vs time starts fluctuating around a constant value after about 80 ps. The red dashed line corresponds to the average total energy from the last 50 ps.

We believe we have addressed the second part of the question in response to comment 1, particularly in the discussion of critical stress, which is a measure of the highest barrier. It is conceivable that the dissociation distance at 850K increases if the simulation is run much longer, but again, this points to the fact that a thermally activated force is acting on the partial dislocation in addition to the elastic force and SFE. Also, note that the TEM observations are at low temperature. In fact, we went as high as 850K precisely to check if thermal energy could completely decorrelate the partial dislocations at a shorter time compared to the low temperature case, and it did not.

Comment 6: “Small typos:

Throughout the paper: Franks rule -> Frank’s rule

Page 3: These questions becomes -> These questions become

Page 7: “staking fault” -> stacking fault”

Response: We thank the reviewer for a careful reading of our manuscript. These typos are fixed.

Reviewer #2

Comment 0: “The manuscript tries to answer the question of the legitimacy of using average SFE as a representative quantity for describing the dislocation structures in HEA. To this end, the authors did a careful study of the local chemical environment's effect on the structure of the dislocations in NiCo alloy. However, I have some concerns about the publication of this paper in NCOMMS at its current form.”

Response: We appreciate the reviewer’s thorough review and helpful suggestions.

Comment 1: “The authors only used the isotropic elasticity as their theoretical framework. I strongly suggest the authors modify their formulation to the case of anisotropic elasticity. If it is not possible to find the closed form solution for the separation distance versus the character angle, they should at least try to use anisotropic elasticity for screw and edge cases and discuss the possible effects.”

Response: We thank the reviewer for raising this point. We have redone all our analyses using anisotropic elasticity and updated the manuscript accordingly. These changes are reflected in equation 1 and Figure 1(b). We also added more details in supplementary note I. There are some changes as discussed in the text, but the major conclusions remained unchanged.

Comment 2: “The authors didn't consider the temperature dependence of the elastic moduli in this work. I guess at 850K the elastic moduli are considerably less than their 0K values. This decrease will change F_{elastic} . For coming up any sound conclusion about the separation distance, we need to analyze the whole temperature-dependent parameters. Sidenote: Even the Burgers vector magnitude will increase by increasing the temperature, but I guess its effect would be minimal. The anisotropy effect may intensify/decrease by increasing the temperature.”

Response: Thanks for pointing out the incompleteness in our studies. We computed the temperature-dependent elastic constants and updated our results, as follows. We used the strain fluctuation formalism under finite temperature to calculate the elastic constants at different temperatures:

We have added more details in the supplementary note II and updated Figure 5 with the new F_{elastic} lines. Using the finite temperature elastic constants obtained, a comparison between the anisotropic at 0K and 850K is shown below (blue and red curves). The gray curve is the original value of F_{elastic} before the revisions. (“5NN” vs “array” highlights the changes made in response to the next comment).

Note that for the lattice constant, the averaged thermal expansion ratio was 1.0095 at 850 K. There is some reduction in the elastic force at finite temperature, but the difference was not big enough to change our argument. We have modified Figure 5 and included both the original and finite temperature elastic constant results. To better compare the lines, we have also split the original figure into three subfigures based on alloy composition to decrease the complexity of the plot. In addition, a brief discussion of the temperature effects on $F_{elastic}$ is added on p.9, line 207-211.

Comment 3: “For Eq. 2, the authors only considered the five nearest dislocations. I guess one can include all the neighbors and find the close form of the F_{elas} and then use it in Fig. 5. The series looks convergent.”

Response: Thanks for the suggestion! We have reevaluated the closed form of $F_{elastic}$ using anisotropic elasticity and considering an array of dislocations as suggested. We have modified equation 2 and Figure 5 accordingly. The series is indeed convergent, and the closed-form solution is:

$$F_{elastic} = \frac{Kb_i b_j}{2\pi} \left(\frac{1}{d_{ij}} + \frac{-L_x + \pi d_{ij} \cot(\pi d_{ij}/L_x)}{L_x d_{ij}} \right)$$

More details have been added to the Supplementary note I.

Comment 4: “The authors used the difference in the free energy of hcp and fcc phases to prove that the SFE remains almost constant in the interest temperature domain. This approximation is fine. However, one can improve it by using the hcp double-layer energies [1]. I suggest the authors consider this improvement in their approximation at least for some representative samples.”

[1] Zhang, Xi, et al. "Temperature dependence of the stacking-fault Gibbs energy for Al, Cu, and Ni." *Physical Review B* 98.22 (2018): 224106.

Response: We agree with the reviewer that including dchp layers improves this model and did the additional calculations as below. The first and second-order axial-next-nearest neighbor-ising (ANNNI) model suggests [1]:

$$\gamma_1(P, T) = \frac{2[F_{\text{hcp}}(V_{\text{fcc}}, T) - F_{\text{fcc}}(V_{\text{fcc}}, T)]}{A_{\text{fcc}}(T)}, \quad (5)$$

$$\gamma_2(P, T) = \frac{F_{\text{hcp}}(V_{\text{fcc}}, T) + 2F_{\text{dhcp}}(V_{\text{fcc}}, T) - 3F_{\text{fcc}}(V_{\text{fcc}}, T)}{A_{\text{fcc}}(T)}, \quad (6)$$

Based on the equations, the estimated SFE using first and second order ANNNI model for NiCo alloys, as well as pure elements:

For the NiCo alloys in our study, the SFE from the two models are almost identical. Hence, Figure 4(a) remains a reasonable estimation for this potential. We have added the discussions in the Supplementary note II, line 90.

Comment 5: "The notion of the friction force is vague. Given the complete randomness of the system, one probably could quantify this friction force in terms of the fluctuations in the atoms' spatial distributions. This analysis should be the same as Varvenne et al. model for solute strengthening in HEA. The difference would be that now we have multiple partial dislocations repelling each other instead of the one dislocation under a resolved shear."

Response: We intentionally used “friction force” to keep it more general than solute strengthening. But we agree that our original presentation of the friction force has not been sufficiently clear. We have added a new figure (Figure 7) and discussions to the text (page 12, lines 260--276) to clarify that.

In principle, we agree that the Varvenne et al. method can be used as a framework to estimate this type of interactions. But there need to be modifications. First, as discussed in response to reviewer #1’s first comment, the problem we are studying is not exactly solute strengthening, although there are similarities in the nature of dislocation/solute interaction. We are studying the dissociation of a lattice dislocation under zero stress. While the partials dislocations repel one another, the entire dislocation is not moving and thus not bowing out. Varvenne et al, assume a perfect, undissociated dislocation, which is reasonable since the length scale of dislocation bowing segments are generally much larger than the dissociation distance between partials. It is also assumed that the two partials are always moving together. In order to apply the analysis to partial dislocations, the effects of solutes on SFE and the correlation with the other partial dislocation should be incorporated into the model. These problems are nontrivial for the case of very low (negative) SFE and partials that can decorrelate. In fact, these issues are the main questions we tackle in this paper. Making any assumptions about them would result in a circular argument. Therefore, we decided to isolate the problem of strengthening from these smaller-scale effects. Hopefully, our results will provide insight to make physical assumptions for the generalization of the Varvenne method for alloys with very low SFEs.

Comment 6: “After implementing the major revisions suggested here, I think manuscript would be suitable for the publication in NCOMMS.”

Response: Once again, we thank the reviewer for their helpful suggestions and believe that the extensive revisions to the manuscript to address both reviewers’ concerns have made the manuscript stronger.

REVIEWER COMMENTS

Reviewer #1 (Remarks to the Author):

The authors have addressed all my comments and hence I suggest that the paper is accepted for publication. I truly appreciate the detailed rebuttal.

Reviewer #2 (Remarks to the Author):

The authors significantly revised the manuscript and answered my concerns about the details of the simulations. However, given the yet vague definition of the friction force, I am a little bit not convinced to accept the manuscript for NATCOMM in its current form.

1-I strongly suggest the authors include the approximate periodic energy landscape as given in Varvenne et al. (Acta Mat. 2016) into account and analyze the enhancement in their model by adding this modification to their theory. The authors have all the pieces. They have elastic repulsion and SFE. The only missing piece is the energy landscape that a partial dislocation feels from the solute distribution. The authors should follow the explanation given right before Eq. (8) of the aforementioned paper. And then they will have an additional force. Having access to that force we can see this first-order approximation is good or not.

I am aware of the differences between the current problem and the Varvenne et al. problem. However, I guess including this first-order approximation enhances the paper and fills the gap between the experimental and theoretical part of the paper which is dedicated to different alloys.

2- It seems that in Fig. 3 and for the Co85Ni15 case, the temperature doesn't significantly change the separation distance. Why is that? Since we don't have the SFE of this alloy at this temperature. I cannot figure out the reason for this anomaly in the T-dependence of the separation distance. This point should be clarified. And I guess this clarification will help the finding the origin of the friction force. It seems that the friction force here has a strong effect. On the other hand, the distance at 850 K is smaller than 78K which even defeats the concept of friction.

I guess the publication of this manuscript in NATCOMM will be possible when the authors can more clarify the origin of the friction force and rationalize the temperature and concentration dependence of it.

Reviewer #1:

Comment 0: “The authors have addressed all my comments and hence I suggest that the paper is accepted for publication. I truly appreciate the detailed rebuttal.”

Response: We thank the reviewer for a thorough review of our work and their favorable opinion of the work.

Reviewer #2:

Comment 0: “The authors significantly revised the manuscript and answered my concerns about the details of the simulations. However, given the yet vague definition of the friction force, I am a little bit not convinced to accept the manuscript for NATCOMM in its current form.”

Response 0: We truly appreciate the reviewer for pushing us to clarify the origin of the “friction” force. As explained below, we now provide a better understanding of forces acting on the Shockley partials during dissociation and have replaced the vague term of “friction” with operative mechanisms such as solute/dislocation interaction or maximum stacking fault energy.

Comment 1: “I strongly suggest the authors include the approximate periodic energy landscape as given in Varvenne et al. (Acta Mat. 2016) into account and analyze the enhancement in their model by adding this modification to their theory. The authors have all the pieces. They have elastic repulsion and SFE. The only missing piece is the energy landscape that a partial dislocation feels from the solute distribution. The authors should follow the explanation given right before Eq. (8) of the aforementioned paper. And then they will have an additional force. Having access to that force we can see this first-order approximation is good or not.

I am aware of the differences between the current problem and the Varvenne et al. problem. However, I guess including this first-order approximation enhances the paper and fills the gap between the experimental and theoretical part of the paper which is dedicated to different alloys.”

Response: We thank the reviewer for this suggestion; we have included the approximation of solute effects on the partial dislocation core in pp. 12-14, including new Figures 7 and 8, and Supplementary Section 4. Based on the work of Varvenne et al., we approximated the solute/dislocation interaction ($\Delta E'_b(L_z, w)$) using two different methods-- the reduced elastic interactions model and directly calculating the interaction energies of Ni solutes in fcc Co with the MEAM potential. Since we consider a straight dislocation without any bow-out, there is no critical roughening amplitude (w_c). Instead, the energy barrier depends on the dislocation line length (L_z) and gliding distance (w). We then define a force per unit dislocation line exerted by the solutes as

$$F_s = \frac{\Delta E'_b(L_z, w)}{wL_z}$$

The maximum value of F_s is compared to the decorrelation force and stacking fault energy in Fig.6 in each alloy.

Comment 2: “It seems that in Fig. 3 and for the Co85Ni15 case, the temperature doesn't significantly

change the separation distance. Why is that? Since we don't have the SFE of this alloy at this temperature. I cannot figure out the reason for this anomaly in the T-dependence of the separation distance. This point should be clarified. And I guess this clarification will help the finding the origin of the friction force. It seems that the friction force here has a strong effect. On the other hand, the distance at 850 K is smaller than 78K which even defeats the concept of friction.”

Response: We again thank the reviewer for insisting on clarifying the origin of the “friction” force. When we initially wrote the manuscript, our intent was to show that such additional resistance exists and that it must be considered. We were not planning to delve into its nature. But, with the help of both reviewers, we now have different pieces of the total resistance and can put together the complete picture.

In the original calculations, the $Ni_{85}Co_{15}$ and $Ni_{90}Co_{10}$ simulations were started with two partial dislocations at an initial finite separation distance, instead of a full dislocation. We had done this to check if the partial dislocations still stopped from completely decorrelating, hence showing the existence of a resistance that cannot be explained by the average SFE. Therefore, the final separation distances at 78K were larger than what they should have been. We agree with the reviewer that this was misleading.

To fix this issue, we have rerun all the simulations starting from a full dislocation dissociating to the equilibrium separation distance for all 3 alloy cases ($Ni_{70}Co_{30}$, $Ni_{85}Co_{15}$, and $Ni_{90}Co_{10}$). In addition, we have defined a convergence criterion for the separation distances to determine when to stop the simulations.

The new results are presented in Fig. 3. We still obtain finite dislocation dissociation distances at 78 K for all three alloys and at 850 K for $Ni_{70}Co_{30}$ and $Ni_{85}Co_{15}$ alloys. However, the partial dislocation decorrelated for the $Ni_{90}Co_{10}$ alloy case at T=850K with the new criterion. The dissociation distances at various temperatures do not show any apparent T-dependence anomaly anymore.

We have recalculated our decorrelation forces and modified all the figures that are affected by this change. Our initial conclusion about the existence of additional resistance and the inadequacy of SFE alone to account for balancing the elastic interaction remains unchanged.

Comment 3: “I guess the publication of this manuscript in NATCOMM will be possible when the authors can more clarify the origin of the friction force and rationalize the temperature and concentration dependence of it.”

Response: Putting everything together, Figures 5 and 6 explain the process as follows. The equilibrium separations (in the absence of any thermal energy) are shown by circles in Fig.5 and intersect the $F_{elastic}$ curves at values corresponding to the decorrelation forces computed independently in Fig.6. In addition, Fig.6 reveals that the decorrelation force in each alloy is equal to the maximum of SFE (γ^{max}) and F_s which are computed separately and independently.

Note that (γ^{max}) and ($F_s = wL\Delta E'_b$) represent barriers to the motion of the partial dislocation and do not change with temperature themselves. At finite T, the dislocation has some help from thermal energy so it can overcome the barrier at lower stress. The two extreme cases are T=0 which gives the maximum stress (force) required to overcome the barrier and T larger than some critical value where the barrier

can be overcome with thermal energy alone and no stress. Note that, we have defined the decorrelation force as the maximum critical value, representing the $T=0$ extreme. Quantitative modeling of a thermally activated process at finite T with MD is problematic because it requires very long time.

We have deleted the ambiguous term “friction” throughout the manuscript and have explained the origin of various forces resisting the complete separation of partial dislocations. We have added new text [page 15, lines 335-353] and revised Fig 9 to explain the process.

REVIEWERS' COMMENTS

Reviewer #2 (Remarks to the Author):

The authors have revised the manuscript satisfactorily and answered all of my questions and addressed all of my previous concerns.

There is only one very minor issue with the definition of the F_s which is approximated by Eq. 4 in the manuscript. Why the authors didn't use the differentiation w.r.t w rather than dividing the energy by the distance w ? I guess the authors can add a sentence discussing this approximation and then the manuscript can be published without any other impediment. I ask the respected editor to handle this very minor issue.

Best regards,

Reviewer #2 (Remarks to the Author):

Comment 1: “The authors have revised the manuscript satisfactorily and answered all of my questions and addressed all of my previous concerns.

There is only one very minor issue with the definition of the F_s which is approximated by Eq. 4 in the manuscript. Why the authors didn't use the differentiation w.r.t w rather than dividing the energy by the distance w ? I guess the authors can add a sentence discussing this approximation and then the manuscript can be published without any other impediment. I ask the respected editor to handle this very minor issue.”

Response: We have added a sentence before equation 4, stating that we used this approximation for simplicity.